# Preliminary Psychometric Evaluation of Novel Measures of Therapist Practice Related to LGBTQ+ Clients

**DOI:** 10.3390/healthcare12010110

**Published:** 2024-01-03

**Authors:** Rodman Turpin, Jessica N. Fish, Evelyn King-Marshall, Bradley Boekeloo

**Affiliations:** 1Department of Global and Community Health, College of Public Health, George Mason University, Fairfax, VA 22030, USA; 2University of Maryland Prevention Research Center, College Park, MD 20742, USA; jnfish@umd.edu (J.N.F.); eckm@umd.edu (E.K.-M.); boekeloo@umd.edu (B.B.); 3Department of Family Science, School of Public Health, University of Maryland, College Park, MD 20742, USA; 4Department of Behavioral and Community Health, School of Public Health, University of Maryland, College Park, MD 20742, USA

**Keywords:** mental health, therapists, sexual minority, gender minority, services, measurement

## Abstract

Background: Culturally competent and equitable mental healthcare for LGBTQ+ people is critical for addressing mental health inequities for this population. Tools to assess therapists’ practice with LGBTQ+ clients are needed for research and clinical efforts related to mental healthcare equity goals. Methods: We conducted a preliminary assessment of the reliability and validity of a novel 28-item self-report measure assessing therapist practice with LGBTQ+ clients. We examined the construct validity using factor analyses, the convergent and criterion validity using intercorrelations with LGBTQ-affirming knowledge, self-efficacy, and attitudes, and the internal consistency using Cronbach alpha. Results: Our overall total LGBTQ+ practice measure demonstrated excellent internal consistency (Cronbach’s alpha = 0.91) and was strongly associated with LGBTQ+ knowledge (rho = 0.377), self-efficacy (rho = 0.633), and LGBTQ+ attitudes (rho = 0.305). We also identified two subscales: “Commitment to Continued Learning” and “Affirmative Practices”, which demonstrated similarly strong internal consistency and tests of validity. Conclusions: Our novel measure of overall LGBTQ+ practice, including two subscales, demonstrated strong reliability and validity. These findings have important implications for practice and research in mental healthcare for LGBTQ+ clients. Future research exploring these measures in relationship to mental healthcare outcomes is recommended.

## 1. Introduction

LGBTQ+ individuals experience disproportionate mental health disparities, including higher rates of depression, anxiety, traumatic stress, psychological distress, and suicidal thoughts [1,2,3,4]. Several studies have demonstrated that these adverse mental health outcomes are influenced by various factors, such as the stigma and stress that LGBTQ+ people encounter, in line with the minority stress theory [3,4,5,6,7,8]. For example, one study by Lattanner, Pachankis, and Hatzenbuehler demonstrated strong relationships between homophobia-related stressors and depression among gay and bisexual men [3]. According to this theory [7,8], marginalized groups such as LGBTQ+ individuals face significantly greater identity-related oppression and stigmatization than their majority peers, which has detrimental effects on their mental and physical well-being, including the aforementioned adverse outcomes [2,3,7,9,10].

Given these disparities, LGBTQ+ people are more likely to need and engage in mental healthcare but are also less likely to receive satisfactory and competent care [11,12,13,14,15]. A study by Silveri et al. found that LGBTQ+ people experience higher rates of substance misuse and mental illness, and unfortunately, also experience greater stigmatization both within and outside healthcare settings, with few tailored interventions to meet their care needs [12]. This creates substantial barriers in accessing health care services. These barriers limiting LGBTQ+ persons’ access to quality health care extend to mental healthcare, where they are further exacerbated by a lack of culturally sensitive providers [12,14,15,16,17,18,19]. For example, a study by Williams and Fish found that 12.6 percent of mental health and 17.6 percent of substance abuse facilities reported having LGBT-specific programs, suggesting the limited availability of culturally competent mental health and substance abuse treatment [15]. As a result, significant gaps persist in providing equitable care for LGBTQ+ individuals, with notable deficiencies in culturally competent care for various subgroups, such as LGBTQ+ homeless populations and youth [19,20]. This is concerning considering the additional mental health stressors that LGBTQ+ people endure due to homophobia, transphobia, and other forms of stigma and discrimination. Addressing these disparities in mental healthcare is of utmost importance to ensure the well-being of LGBTQ+ individuals.

The measurement of LGBTQ+-affirmative practice is necessary to help (1) assess the degree to which therapists are prepared to work with this population; (2) assess areas for improvement through training and education; and (3) evaluate the effectiveness of programs to change the clinical practice of clinicians working with this population. As such, these measures are essential in identifying growth areas and improving therapist training and infrastructure for LGBTQ+ services. Unfortunately, limited tools are available to assess the services provided by mental healthcare providers, particularly concerning LGBTQ+ clients [21,22,23]. The lack of measures in this area further stymie efforts to advance policies, programs, and practices designed to increase healthcare workforce readiness to serve LGBTQ+ populations. This deficit is increasingly relevant, given the growing number of LGBTQ+ people in the United States and the population’s increasing mental healthcare needs, particularly among young people [15,18,19,24,25]. Therefore, it is crucial to have validated measures that can effectively assess LGBTQ+ mental healthcare services among therapists to better meet health equity needs for this community.

Our study aimed to assess the validity of a novel 28-item self-report measure of affirmative mental healthcare provider practice related to LGBTQ+ clients (Appendix A). We assessed the dimensionality, internal consistency, convergent validity, and criterion validity of the measure. The criterion validity was assessed using knowledge, self-efficacy, and attitude measures related to the affirming care of LGBTQ+ clients, as these concepts have generally been associated with culturally competent care for minoritized populations. If this measure demonstrates validity, it can be further studied for use as a self-assessment by therapists to gauge their own practice and areas for improvement regarding the care of LGBTQ+ clients. Furthermore, this measure can be further studied for use in research to understand the rates of therapists’ affirmative LGBTQ+ practice, compare practice across different groups of therapists, and/or examine changes in therapists’ affirmative LGBTQ+ practice over time.

## 2. Materials and Methods

### 2.1. Sample

We utilized data from a randomized controlled trial (RCT) of mental health organizations conducted by the University of Maryland Prevention Research Center. Data collection took place from January 2021 to June 2022. The trial’s main objective was to enhance the competency of community mental health organizations and their therapists in providing mental healthcare to LGBTQ+ individuals. Therapists were recruited by their organizations’ leaders (e.g., Executive Directors, Clinical Directors). To be eligible for participation, therapists needed to meet specific criteria, including being provisionally or fully licensed therapists (such as clinical social workers, mental health counselors, licensed professional counselors, licensed psychologists, and licensed marriage and family therapists), working at least 20 h per week at the organization, and having a minimum of 10 clients aged 16 or older. A total of 48 therapists completed all the surveys (See Table 1 for their demographic characteristics). The study procedures were approved by the institutional review board of an undisclosed institution.

### 2.2. Measures

Measures were developed by a large and diverse research team including practicing therapists, behavioral and mental health researchers with and without clinical training, and public health students and faculty who were members of the LGBTQ+ community. Additionally, a Community Advisory Board (CAB) consisting of various LGBTQ+ stakeholder groups (researchers, therapists, policy makers, and LGBTQ+ community members) who represented diverse races, ethnicities, sexual orientations, and gender identities reviewed and provided input throughout the development of the measures. First, a list of 20 therapist competencies for affirming care of LGBTQ+ clients was developed. Based on a literature review of existing measures and using the competencies as a guide, the research team and CAB drafted a survey across four domains: knowledge, attitudes, self-efficacy, and practice related to affirming care of LGBTQ+ clients, and then conducted multiple rounds of review and revision. Based on this exhaustive process, we identified knowledge, attitudes, self-efficacy, and LGBTQ+-affirming care as key criterion domains; these criterion domains are described in more detail in the parent study [26]. The final Therapist LGBTQ+ Competence Self-Assessment was administered to therapists through the Qualtrics online platform at study baseline before therapists were randomized to study condition and study intervention as part of the larger RCT.

The therapists responded to 28 Likert scale items that assessed different aspects of their practice towards LGBTQ+ individuals and their mental healthcare. The items included statements such as “I stay current with the language used by LGBTQ+ people”, “I help clients with gender dysphoria embrace their gender identity”, and “I help LGBTQ+ clients identify their own internalized homophobia, biphobia, and/or transphobia”. All items were measured using a Likert scale measured as 0 (Strongly Disagree), 0.25 (Disagree), 0.5 (Neutral), 0.75 (Agree), and 1 (Strongly Agree). Items that required reverse coding were reversed before conducting any analyses.

We also examined various therapist demographic factors such as gender (cisgender woman, cisgender man), age groups (21 to 30, 31 to 40, 41 to 50, 51 or older), race (Asian/Pacific Islander, Black, White), ethnicity (Hispanic/Latino, Non-Hispanic/Latino), and sexual orientation (asexual, bisexual, gay/lesbian, heterosexual). It is worth noting that none of the participants identified as transgender or nonbinary, despite available options. Socioeconomic status measures, including household income and education level, were not included since they were similar among all the therapists.

### 2.3. Missing Data and Data Quality

Missingness across all variables was very low, with less than 6% missingness for all items, so we used intrascale stochastic imputation to impute missing values. It is appropriate when missing values occur randomly and there is sufficient internal consistency among items, as we determined was consistent with our missing data [27]. All non-missing practice items were utilized in imputing missing values. Although this approach creates marginally more overall consistency, it does not substantially impact our ability to identify subscales since there are so few missing values in the unimputed dataset. For analyses, leverages and Cook’s distances were both used to assess outliers; no observations demonstrated unusually high Cook’s distances or leverages.

### 2.4. Analyses

We first conducted an exploratory factor analysis with maximum likelihood factoring and varimax rotation (Boateng et al., 2018) to identify potential subscales. Only items with a factor loading greater than 0.40 were retained as part of that factor. After examining the factors, we conducted sensitivity analyses using post-test data to determine if the factor structure remained consistent (>90% agreement in item grouping) between the baseline and post-test assessments [27] to identify potential subscales. Subsequently, we created subscales by summing the items with the highest correlation with each identified factor. We also included an a priori summative subscale consisting of only the items specific to clients’ gender identity, as this subscale was of particular interest regardless of the factor structure. The unreduced and reduced total scale and subscales were included in all convergent and criterion validity analyses.

The items included in the summative scales, which were developed based on the factor structure analysis mentioned above and in the scale related to client gender identity, were analyzed to evaluate their contributions to the internal consistency of each scale. We examined how well each item correlated with the overall scale by calculating item correlations with the scale’s total score. Additionally, we assessed the Cronbach alpha coefficient of the scale with each item removed to determine the impact of each item on the internal consistency of the overall scale. Items that appeared to decrease the overall internal consistency were removed from the summative scale. Cronbach alphas of 0.70 or greater were considered acceptable.

To evaluate the convergent validity of the subscales, we examined the intercorrelations between each practice subscale. We expected that the different subscales would demonstrate positive correlations with each other since they should all capture different aspects of LGBTQ+ practice. The absence of correlation could suggest distinct and independent subdomains of practice. We assessed criterion validity by examining the correlations between the practice measures and the knowledge, attitude, and self-efficacy measures, employing Spearman’s rank-sum correlation coefficients for the analysis [28]. We conducted all analyses in SAS 9.4 [29].

## 3. Results

### 3.1. Sample Characteristics and Reported Practices

Our sample consisted of 48 participants (Table 1). The majority of the participants were cisgender women (87.5%), white (72.9%), and non-Hispanic/Latino (91.7%). Just over a fifth (22.9%) of participants were sexual minorities, and all therapists were cisgender. The participants were diverse across the age groups. The participants generally endorsed practice items in an LGBTQ+-affirmative way (Table 2), as the mean score of 19 of the 28 items leaned toward frequent practice, with a mean greater than 0.60 on a 5-point scale as follows: Never (0), Almost never (0.25), Sometimes (0.5), Frequently (0.75), All the time (1). The nine exceptions were the items regarding staying connected with LGBTQ+ resources (mean = 0.50), supporting clients who want gender confirmation surgery (mean = 0.58), sharing pronouns when introducing oneself (mean = 0.32), asking consent before using sexual orientation and gender identity language (mean = 0.48), asking consent before using any potentially sensitive language (mean = 0.50), helping LGBTQ+ clients identify their own internalized homophobia, biphobia, and/or transphobia (mean = 0.58), helping clients identify external sources of internalized homophobia, biphobia, and/or transphobia (mean = 0.59), having a list of LGBTQ+ support services (mean = 0.29), and staying abreast of laws protecting LGBTQ+ rights (mean = 0.44).

### 3.2. Dimensionality and Internal Consistency

Our factor analysis identified two factors that explained most of the variance of the items (Table 3). Factor 1 consisted of 19 items broadly related to LGBTQ+-affirmative practices. Factor 2 consisted of 8 items, most primarily centered around commitment to continued learning related to the mental healthcare of LGBTQ+ clients. As described previously, we also included an a priori face-valid scale of five items focused on the gender-affirmative practice items, given the uniqueness and importance of these skills. For these three subscales, each group of items was reduced, removing any items that when removed, increased the internal consistency of each scale. This resulted in a reduced “Commitment to Continued Learning” subscale of four items, a reduced “LGBTQ+-Affirmative Practices” subscale of nine items, and a reduced “Gender-specific Affirmative Practice” subscale of three items (Table 4). All subscales demonstrated an acceptable internal consistency (Cronbach’s alpha > 0.70), except the unreduced “Gender-Affirmative Practice” subscale, though the reduced version did have a modestly sufficient internal consistency (Cronbach’s alpha > 0.74). The reduced versions of all four subscales had a higher absolute internal consistency than the unreduced versions.

### 3.3. Convergent and Criterion Validity

Because all of the reduced versions of the subscales had a higher internal consistency than the unreduced versions, only their correlations with other variables in the convergent and criterion validity analysis are presented. Without exception, these results mirrored, albeit usually with slightly more association strength, the results with the unreduced subscales. All four reduced subscales were positively associated with each other (Table 5). All four reduced subscales were also positively associated with self-efficacy. With the exception of the reduced Gender-Affirmative Practice scale, the other three subscales were positively associated with knowledge and attitudes.

## 4. Discussion

The LGBTQ+ practice scale consisted of two differentiated subscales regarding “LGBTQ+-Affirmative Practice” and “Commitment to Continued Learning”. Shortened versions of these subscales had a higher internal consistency (reliability) than the initial longer versions. Given the potential advantages of shorter assessments in regard to participant time and effort, the more reliable shorter subscales were further examined and found to correlate with each other and with other criterion variables in ways that increased the confidence that the separate subscales, and the subscales combined into one scale, are valid in regard to the measurement of the therapists’ competence with LGBTQ+ clients. The positive relationships of both subscales and overall combined measure with the criterion measures confirmed the intuitive hypotheses and were supported by the literature, as affirmative practices towards LGBTQ+ people, including culturally competent care, are often related to affirmative attitudes, self-efficacy, and knowledge [16,17,19,22,30]. In total, this study suggests that the new measures of LGBTQ+-affirmative practice and commitment to continued learning, as separate subscales or together as one scale, warrant further attention for administration to advance research and practice. These subscales may be administered individually or as one overall scale to assess therapists’ relative level of competence with LGBTQ+ clients, with higher scores interpreted as meaning higher competence. The LGBTQ+-affirmative practice subscale may be particularly useful for assessing current practices related to LGBTQ+ clients, while the “Commitment to Continued Learning” subscale may be a useful tool for capturing potential sustainability of practice and continued growth in LGBTQ+ mental healthcare competency.

Given the importance of gender-affirming practice with clients, affirmative practice items related to clients’ gender identity based on face-validity were also examined independently as a summative subscale. This “Gender-affirmative practice” subscale in the original and shortened forms barely reached an acceptable level of reliability. Furthermore, while this subscale showed correlation with the other subscales as well as LGBTQ+ practice self-efficacy, it did not correlate with the knowledge and attitude measures. This may relate to the small number of items in this subscale; an increased number of items usually increases the internal consistency alpha and improves the predictive power. Furthermore, the survey was not developed with a gender identity-specific subscale in mind; hence, the domains covered by the items may not be adequate. Hence, while such a measure is important for research and practice related to therapist competence with LGBTQ+ clients, the gender-affirmative practice subscale in this study did not demonstrate adequate enough psychometric characteristics to suggest that it is useful for research or practice in its current form.

It was important to note which practices were more frequently reported by therapists versus those that were less frequently reported. Therapists generally reported frequently using clients’ pronouns, helping to build their resources, normalizing their feelings, helping them build social support, and not assuming that their reason for seeking therapy was due to their sexuality or gender. Other practices were notably infrequently reported, however, such as sharing their own pronouns, staying connected to LGBTQ+-related professional development opportunities, seeking client consent around language relating to sex, sexuality, and gender, keeping a list of affirming health and social service resources, and staying abreast of LGBTQ+-targeted legislation that may significantly impact mental health and well-being. The former practices may be in-keeping with generic counseling skills taught in most therapist training programs. The latter practices may be less frequently encountered by therapist training programs and only taught in more specialized training related to sexuality and gender. This suggests that therapists need more of this specialized training related to sexuality and gender, that such training needs to be on-going given that sociopolitical contexts affecting the mental health of LGBTQ+ persons are constantly evolving, and that overall, more therapist training programs need to address competence regarding the topics of sex, sexuality, and gender.

Validated measures of LGBTQ+-affirmative practice are necessary tools in the effort to improve the education and training of mental healthcare providers, given the current gaps in LGBTQ-affirmative services and the current reports of continued harm in mental health practice for LGBTQ+ individuals [15,17,19,25,31,32]. Measures of mental healthcare providers’ readiness and limitations in their work with LGBTQ+ clients are crucial for assessing the degree to which training programs support the development of practices necessary to work effectively with LGBTQ+ clients. These tools also help to advance research in this area. Overall, measures of LGBTQ+-affirmative practice among therapists are needed to identify how therapists can improve their practice and to conduct research regarding the improvement in therapist practice needed to enact health equity goals for this population.

This research has notable strengths. This study explores a novel measure of LGBTQ+-related mental healthcare practice. Developing new methods to assess this holds significance for multiple research inquiries concerning mental health and the disproportionate health disparities experienced by LGBTQ+ individuals. We also utilized a relatively large number of items that address multiple areas of competence regarding the needs of LGBTQ+ clients, enabling the capture of numerous subtle aspects of LGBTQ+ practice among therapists. Finally, factor analyses were employed to identify distinct constructs within the primary practice scale, identifying important multidimensionality.

We must acknowledge certain limitations in our research. The sample of therapists was relatively small, only from Maryland, and only from general, non-specialized mental health organizations. Our factor structure may not replicate exactly in a larger, less targeted sample. This may restrict the findings’ generalizability as a larger, more diverse sample may generate somewhat different findings that better represent a broader pool of therapists. Similarly, the therapists in this study had high rates of LGBGTQ+-related knowledge, so it is possible that the sample, which was also motivated to participate in the trial, had generally higher LGBTQ+-specific competency than therapists at large. The measures tested in this study are newly developed by the research team and have not yet undergone repeated reliability and validity examination in different samples of therapists. Finally, the measures are based on self-report with potential related biases, including social desirability bias related to sensitive questions [33]. For these reasons, this study should be considered as preliminary and in need of further replication before the results can be considered conclusive. Nevertheless, despite this limitations, the identified statistical results, especially the associations between our total practice scale, subscales, and criterion measures, appeared quite strong, increasing our confidence in their integrity.

## 5. Conclusions

Our novel measure of therapists’ self-reported practice with LGBTQ+ clients demonstrated important dimensions with strong overall validity, including strong internal consistency and hypothesized relationships with LGBTQ+ knowledge, self-efficacy, and LGBTQ+ attitudes. The overall measure revealed two subscales, one focused on commitment to continued learning and the other on LGBTQ+-affirmative practice. The overall measure and two subscales performed particularly well in the tests of validity after deletion of internally inconsistent items, creating relatively short measures that minimize therapist burden, making them conducive to many research and practice applications. Our findings have important implications for practice and research related to LGBTQ+ mental healthcare. Future research exploring the associations between these measures and LGBTQ+ mental healthcare outcomes is recommended.

## Figures and Tables

**Table 1 healthcare-12-00110-t001:** Baseline sociodemographic characteristics of therapists (n = 48).

	Total %
**Age (years)**	
21 to 30	22.9
31 to 40	37.5
41 to 50	20.8
Over 50	18.8
**Ethnicity**	
Hispanic/Latino	8.3
Non-Hispanic/non-Latino	91.7
**Race**	
Asian/Pacific Islander	6.3
Black	20.8
White	72.9
**Sex Assigned at Birth**	
Female	87.5
Male	12.5
**Gender**	
Cisgender Man	12.5
Cisgender Woman	87.5
Non-Binary	0.0
Transgender Man	0.0
Transgender Woman	0.0
**Sexual Identity**	
Asexual	2.1
Bisexual	12.5
Gay/Lesbian	8.3
Heterosexual	77.1

**Table 2 healthcare-12-00110-t002:** Means and standard deviations for practice items (n = 48).

	Mean	Standard Deviation
1. I engage in a process of self-reflection to assess my own attitudes and emotions towards LGBTQ+ clients.	0.698	0.199
2. I stay current with the language used by LGBTQ+ people.	0.635	0.186
3. I stay connected with LGBTQ+ resources for professional development related to LGBTQ+ competency.	0.500	0.213
4. I help clients explore the meaning of their physical sexual drives, romantic attractions, and actual sexual behaviors when they are questioning their sexual orientation.	0.646	0.291
5. I help clients with gender dysphoria embrace their gender identity.	0.630	0.236
6. I support clients who want gender confirmation surgery in obtaining the affirmative healthcare that they need.	0.578	0.319
7. I infer clients’ sexual orientation based on their gender identity (Reversed).	0.844	0.190
8. I infer clients’ gender identity based on their sexual orientation (Reversed).	0.854	0.199
9. I use the name that my client uses regardless of their legal name.	0.969	0.098
10. I share my own pronouns when I introduce myself to clients.	0.318	0.291
11. I use the pronouns that my client uses.	0.896	0.199
12. I ask consent before using the sexual orientation and gender identity language that my client uses.	0.484	0.332
13. I ask consent before using any potentially sensitive language in reference to body parts and/or behaviors that my client uses.	0.500	0.346
14. When I meet a new client, I assess the pronouns and name they use.	0.729	0.277
15. I help LGBTQ+ clients identify and build on their strengths and resources.	0.823	0.219
16. I help LGBTQ+ clients identify their own internalized homophobia, biphobia, and/or transphobia.	0.578	0.278
17. I help clients identify external sources of internalized homophobia, biphobia, and/or transphobia.	0.594	0.256
18. I continually examine my own practice for ways that I might not be supportive to LGBTQ+ clients.	0.630	0.258
19. I try to speak up in my organization when I see things that might demean LGBTQ+ clients.	0.646	0.309
20. I assist my LGBTQ+ clients in obtaining behavioral, social, and medical services when appropriate.	0.724	0.284
21. I have made a list of behavioral, social, and medical services that are supportive of LGBTQ+ persons.	0.286	0.309
22. I do not assume that my LGBTQ+ clients are seeking mental health care because of concerns about their sexuality or gender.	0.828	0.269
23. When desired by LGBTQ+ clients, I explore the impact of racism on their mental health.	0.802	0.273
24. I engage with LGBTQ+ clients’ around developing close bonds with supportive persons in their lives.	0.859	0.162
25. I encourage parents to work through their feelings and support their LGBTQ+ children as they are.	0.760	0.263
26. I normalize LGBTQ+ clients’ feelings during different points of the coming out process.	0.802	0.236
27. I help LGBTQ+ clients navigate their lack of safety related to the coming out process.	0.719	0.299
28. I make a point of staying abreast of laws protecting LGBTQ+ rights in health care, employment, adoption, etc.	0.438	0.239

Response options are on a 5-point scale as follows: Never (0), Almost never (0.25), Sometimes (0.5), Frequently (0.75), All the time (1).

**Table 3 healthcare-12-00110-t003:** Exploratory factor analysis and practice item factor loadings for (n = 48).

	Factor 1 *	Factor 2 *	Factor 3 *
1. I engage in a process of self-reflection to assess my own attitudes and emotions towards LGBTQ+ clients.	0.36927	**0.55883**	−0.29264
2. I stay current with the language used by LGBTQ+ people.	0.09769	**0.63088**	−0.40777
3. I stay connected with LGBTQ+ resources for professional development related to LGBTQ+ competency.	0.31064	**0.68417**	−0.20485
4. I help clients explore the meaning of their physical sexual drives, romantic attractions, and actual sexual behaviors when they are questioning their sexual orientation.	**0.76421**	0.06093	0.13381
5. I help clients with gender dysphoria embrace their gender identity.	0.48466	**0.67411**	−0.06672
6. I support clients who want gender confirmation surgery in obtaining the affirmative healthcare that they need.	0.48209	**0.58808**	−0.05965
7. I infer clients’ sexual orientation based on their gender identity (reversed).	0.09241	**0.68820**	0.45803
8. I infer clients’ gender identity based on their sexual orientation (reversed).	0.19616	**0.73605**	0.47970
9. I use the name that my client uses regardless of their legal name.	**0.33662**	−0.32570	−0.48888
10. I share my own pronouns when I introduce myself to clients.	**0.57721**	0.05354	0.36852
11. I use the pronouns that my client uses.	**0.24929**	−0.30699	−0.46124
12. I ask consent before using the sexual orientation and gender identity language that my client uses.	**0.51292**	−0.26014	0.65969
13. I ask consent before using any potentially sensitive language in reference to body parts and/or behaviors that my client uses.	**0.64281**	−0.09682	0.58835
14. When I meet a new client, I assess the pronouns and name they use.	**0.66766**	−0.25161	0.11424
15. I help LGBTQ+ clients identify and build on their strengths and resources.	**0.61378**	0.08132	−0.08284
16. I help LGBTQ+ clients identify their own internalized homophobia, biphobia, and/or transphobia.	**0.74490**	−0.41196	−0.28592
17. I help clients identify external sources of internalized homophobia, biphobia, and/or transphobia.	**0.76223**	−0.45584	−0.27614
18. I continually examine my own practice for ways that I might not be supportive to LGBTQ+ clients.	0.39378	**0.75892**	0.17734
19. I try to speak up in my organization when I see things that might demean LGBTQ+ clients.	**0.86089**	0.04958	0.07185
20. I assist my LGBTQ+ clients in obtaining behavioral, social and medical services when appropriate.	**0.76473**	0.16999	0.37397
21. I have made a list of behavioral, social, and medical services that are supportive of LGBTQ+ persons.	**0.52288**	−0.12534	0.57932
22. I do not assume that my LGBTQ+ clients are seeking mental health care because of concerns about their sexuality or gender.	0.08408	−0.09675	**0.40892**
23. When desired by LGBTQ+ clients, I explore the impact of racism on their mental health.	**0.64051**	0.45978	0.10189
24. I engage with LGBTQ+ clients’ around developing close bonds with supportive persons in their lives.	**0.61921**	0.64494	0.06297
25. I encourage parents to work through their feelings and support their LGBTQ+ children as they are.	**0.74714**	0.39560	−0.22401
26. I normalize LGBTQ+ clients’ feelings during different points of the coming out process.	**0.69664**	0.17120	−0.50566
27. I help LGBTQ+ clients navigate their lack of safety related to the coming out process.	**0.87467**	−0.07562	−0.12768
28. I make a point of staying abreast of laws protecting LGBTQ+ rights in health care, employment, adoption, etc.	**0.69696**	−0.27028	0.18764

All items with factor estimates >0.40 are emboldened. Varimax rotation was utilized. * Factor 1 items were used to generate the “Affirmative Practices”, while Factor 2 items were used to generate the “Commitment to Learning” subscale. No subscales were generated from Factor 3 due to the dearth of mapped items.

**Table 4 healthcare-12-00110-t004:** Subscales and standardized Cronbach’s alphas for practice items based on factor analysis (n = 48).

	Unreduced		Reduced	
	Items	Alpha	Items	Alpha
Factor 1 Subscale: Affirmative Practices	4, 9, 10, 11, 12, 13, 14, 15, 16, 17, 19, 20, 21, 23, 24, 25, 26, 27, 28	0.90	15, 16, 17, 19, 20, 21, 24, 26, 27	0.92
Factor 2 Subscale: Commitment to Continued Learning	1, 2, 3, 5, 6, 7, 8, 18	0.74	1, 2, 3, 18	0.86
Face Valid Subscale: Gender-Affirmative Practices	5, 6, 9, 10, 11	0.59	5, 6, 10	0.74
Total Practice Scale	1–28	0.84	1, 2, 3, 15, 16, 17, 18, 19, 20, 21, 24, 26, 27	0.91

Reduced subscales omit any items that increased the overall Cronbach’s alpha when they were removed. The shown reduced total practice scale consists of all items from reduced practice subscales. The unreduced gender-specific affirmative subscale items were selected a priori based on face validity.

**Table 5 healthcare-12-00110-t005:** Spearman’s rank sum correlations between practice scales and reduced subscales and criterion measures (n = 48).

	Total Reduced Practice Scale	Reduced Subscale: Commitment to Continued Learning	Reduced Subscale: Affirmative Practices	Reduced Subscale: Gender-Specific Affirmative Practices	Knowledge Scale	Efficacy Scale	Attitudes Scale
Total Reduced Practice Scale	-	**0.842**	**0.976**	**0.666**	**0.377**	**0.633**	**0.305**
Reduced Subscale: Commitment to Continued Learning	-	-	**0.720**	**0.551**	**0.403**	**0.574**	**0.344**
Reduced Subscale: Affirmative Practices	-	-	-	**0.639**	**0.326**	**0.617**	**0.272**
Reduced Subscale: Gender-Affirmative Practices	-	-	-	**-**	0.115	**0.412**	−0.030

Estimates with *p* < 0.05 bolded.

## Data Availability

Data are available upon request. Please contact boekeloo@umd.edu regarding data requests. Additionally, our LGBTQ+ Competency Self-Assessment questionnaire, which includes our practice and criterion measures, is available in the included Appendix A and at the following link: (https://doi.org/10.13016/dspace/tana-axvx).

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
