# Peer review of "Preliminary Psychometric Evaluation of Novel Measures of Therapist Practice Related to LGBTQ+ Clients"

_healthcare, 2024, doi:10.3390/healthcare12010110_

Round 1

Reviewer 1 Report

Comments and Suggestions for Authors

This paper presents a topic of great interest and importance. The lack of assessment tools in the literature to evaluate therapists' practice with LGBTQ+ people is evident. Therefore, I believe that the authors’ effort to build such a tool from the beginning is praiseworthy. The results reported are encouraging from a psychometric point of view; the methodology by which the work was conducted is correct. However, I feel constrained to highlight a major concern of mine, relating to the smallness of the sample of therapists interviewed.

Therefore, I wanted to ask the authors whether they have also considered the use of bootstrapping techniques in applying exploratory factor analysis. I think this additional data could bring strength to this work, which, although preliminary, appears to be very interesting. Clarification on this aspect might be useful.

Author Response

Dear Reviewer, 

We are pleased to submit a revised version of our manuscript entitled “Preliminary Psychometric Evaluation of Novel Measures of Therapist Practice related to LGBTQ+ Clients” for consideration in Healthcare as an original article. We would like to thank all reviewers for their time spent reviewing the manuscript and their detailed feedback. We have revised the manuscript to thoroughly address each of the recommended changes. In our revision response letter, we describe our responses and revisions in bold below each question and recommendation. Quoted comments are revisions added to the manuscript. Sections have also been included in parentheses to indicate areas where revisions begin.

Each of the authors confirms that this manuscript has neither been published nor is simultaneously being considered for publication elsewhere. Each named author has significantly contributed to the underlying research and drafting of this manuscript. The named authors have no conflict of interest, financial or otherwise. I appreciate your consideration and am looking forward to your response.

Reviewer 1

This paper presents a topic of great interest and importance. The lack of assessment tools in the literature to evaluate therapists' practice with LGBTQ+ people is evident. Therefore, I believe that the authors’ effort to build such a tool from the beginning is praiseworthy.

Thank you!

The results reported are encouraging from a psychometric point of view; the methodology by which the work was conducted is correct. However, I feel constrained to highlight a major concern of mine, relating to the smallness of the sample of therapists interviewed. Therefore, I wanted to ask the authors whether they have also considered the use of bootstrapping techniques in applying exploratory factor analysis. I think this additional data could bring strength to this work, which, although preliminary, appears to be very interesting. Clarification on this aspect might be useful.

We appreciate this suggestion and conducted bootstrapping with 1000 repetitions as a post-hoc sensitivity analysis, to determine if the factor structure generated by aggregate bootstrap estimates is similar to our original factor structure. Aggregated bootstrapped estimates were very similar to our original structure, with 92% of items (26 out of 28) mapping onto the same factor-based subscales. After item reduction, subscales generated by bootstrapped estimates and our original approach were identical.

We have added this description to our methods:

“We also conducted bootstrapping with 1000 repetitions as a post-hoc sensitivity analysis, to determine if the factor structure generated by aggregate bootstrap estimates is similar to our original factor structure.”

And the Results:

“Aggregated bootstrapped estimates were very similar to our original structure, with 92% of items (26 out of 28) mapping onto the same factor-based subscales. After item reduction, subscales generated by bootstrapped estimates and our original approach were identical.”

Reviewer 2 Report

Comments and Suggestions for Authors

I appreciate the purpose of this paper and the knowledge gap that it aims to fill. There is significant utility in having a measure of competency in LGBTQ+ mental healthcare. 

The references that describe healthcare disparities and need for competent clinicians are adequate, but could be more extensive.

Overall the methods are clear and conclusions are supported by the interpretation of the data. I would suggest more comment on why you think "gender affirmative practice" subscale didn't show validity.

On line 300, the sentence should finish: LGBTQ+ Affirmative Practice

It could be helpful understand the amount of LGBTQ+ specific training these therapists have had prior to this assessment. Is that information part of the larger data set? This may have implications for generalizability.

Author Response

Dear Reviewer, 

We are pleased to submit a revised version of our manuscript entitled “Preliminary Psychometric Evaluation of Novel Measures of Therapist Practice related to LGBTQ+ Clients” for consideration in Healthcare as an original article. We would like to thank all reviewers for their time spent reviewing the manuscript and their detailed feedback. We have revised the manuscript to thoroughly address each of the recommended changes. In our revision response letter, we describe our responses and revisions in bold below each question and recommendation. Quoted comments are revisions added to the manuscript. Sections have also been included in parentheses to indicate areas where revisions begin.

Each of the authors confirms that this manuscript has neither been published nor is simultaneously being considered for publication elsewhere. Each named author has significantly contributed to the underlying research and drafting of this manuscript. The named authors have no conflict of interest, financial or otherwise. I appreciate your consideration and am looking forward to your response.

Reviewer 2

I appreciate the purpose of this paper and the knowledge gap that it aims to fill. There is significant utility in having a measure of competency in LGBTQ+ mental healthcare. 

Thank you!

The references that describe healthcare disparities and need for competent clinicians are adequate, but could be more extensive.

We have revised to describe this more extensively, in both Paragraph 1 (Introduction, Paragraph 1):

LGBTQ+ individuals experience disproportionate mental health disparities, including higher rates of depression, anxiety, traumatic stress, psychological distress, and suicidal thoughts [1-6]. Several studies have demonstrated that these adverse mental health outcomes are influenced by various factors, such as stigma and stress that LGBTQ+ people encounter, in line with minority stress theory [3-8]. For example, one study by Lattanner, Pachankis, and Hatzenbuehler demonstrated strong relationships between homophobia-related stressors and depression among gay and bisexual men [3].

...and Paragraph 2 (Introduction, Paragraph 2):

“Given these disparities, LGBTQ+ people are more likely to need and engage in mental health care but are also less likely to receive satisfactory and competent care [11-15]. A study by Silveri et al. found that LGBTQ+ people experience higher rates of substance misuse and mental illness, and unfortunately, also experience greater stigmatization both within and outside healthcare settings with few tailored interventions to meet their care needs [12]. This creates substantial barriers in accessing health care services. These barriers limiting LGBTQ+ persons’ access to quality health care extend to mental health care where they are further exacerbated by a lack of culturally sensitive providers [12, 14-19].”

Overall the methods are clear and conclusions are supported by the interpretation of the data. I would suggest more comment on why you think "gender affirmative practice" subscale didn't show validity.

We have revised to describe this further (Discussion, Paragraph 2):

“Furthermore, while this subscale showed correlation with the other subscales as well as LGBTQ+ practice self-efficacy, it did not correlate with the knowledge and attitude measures. This may relate to the small number of items in this subscale--increased number of items usually increases internal consistency alpha and improve the predictive power.  Furthermore, the survey was not developed with a gender identity specific subscale in mind, hence, the domains covered by the items may not be adequate.”

On line 300, the sentence should finish: LGBTQ+ Affirmative Practice.

Thank you, we have revised to correct this typo.

It could be helpful understand the amount of LGBTQ+ specific training these therapists have had prior to this assessment. Is that information part of the larger data set? This may have implications for generalizability.

Measuring LGBTQ+ training is difficult given that various training identified as such may have dramatically different content and quality.  The therapists in this study had high rates of LGBGTQ+ related knowledge according to our measures so it is possible that the sample, which was also motivated to participate in the trial, had generally higher LGBTQ+ specific competency than therapists at-large.  We now identify this in the limitations section (Discussion, Last Paragraph):

“Similarly, the therapists in this study had high rates of LGBGTQ+ related knowledge so it is possible that the sample, which was also motivated to participate in the trial, had generally higher LGBTQ+ specific competency than therapists at-large.” 

Reviewer 3 Report

Comments and Suggestions for Authors

This is a well-written paper that will be very useful to therapists. All of the 28 practice items are what therapists should be following.

Some strengths:

In the discussion section, I like that subscales were examined and potentially can be used. I wish it could go further to suggest to therapists the exact scales and subscales that can be used.

I also like in the discussion section the thoughts on why some practices were given more than others.

Some areas of improvement:

Can you give citations starting on line 104 on the that helped guide the survey to the four domains of knowledge, attitudes, self-efficacy, and LGBTQ+ affirming care? Can you give a statement that these practices were exhaustive enough?

On p. 128, what is "intrascale stochastic imputation" and is there reference citation associated in using that imputation method for missing data?

Author Response

Dear Reviewer, 

We are pleased to submit a revised version of our manuscript entitled “Preliminary Psychometric Evaluation of Novel Measures of Therapist Practice related to LGBTQ+ Clients” for consideration in Healthcare as an original article. We would like to thank all reviewers for their time spent reviewing the manuscript and their detailed feedback. We have revised the manuscript to thoroughly address each of the recommended changes. In our revision response letter, we describe our responses and revisions in bold below each question and recommendation. Quoted comments are revisions added to the manuscript. Sections have also been included in parentheses to indicate areas where revisions begin.

Each of the authors confirms that this manuscript has neither been published nor is simultaneously being considered for publication elsewhere. Each named author has significantly contributed to the underlying research and drafting of this manuscript. The named authors have no conflict of interest, financial or otherwise. I appreciate your consideration and am looking forward to your response.

Reviewer 3

This is a well-written paper that will be very useful to therapists. All of the 28 practice items are what therapists should be following.

Thank you!

Some strengths: In the discussion section, I like that subscales were examined and potentially can be used. I wish it could go further to suggest to therapists the exact scales and subscales that can be used.

We have revised to add recommendations based on specific subscales (Discussion, Paragraph 1):

“These subscales may be administered individually or as one overall scale to assess therapists’ relative level of competence with LGBTQ+ clients, with higher scores interpreted as meaning higher competence. The LGBTQ+ affirmative practice subscale may be particularly useful for assessing current practices related to LGBTQ+ clients, while the “Commitment to Continued Learning” subscale may be a useful tool for capturing potential sustainability of practice and continued growth in LGBTQ+ mental health care competency.”

I also like in the discussion section the thoughts on why some practices were given more than others.

Thank you!

Some areas of improvement: Can you give citations starting on line 104 on the that helped guide the survey to the four domains of knowledge, attitudes, self-efficacy, and LGBTQ+ affirming care? Can you give a statement that these practices were exhaustive enough?

We have revised to add this statement, with citation to the original study, which describes the background for selecting criterion measures in much more detail (Methods, Measures):

“Based on a literature review of existing measures and using the competencies as a guide, the research team and CAB drafted a survey across four domains: knowledge, attitudes, self-efficacy and practice related to affirming care of LGBTQ+ clients and then conducted multiple rounds of review and revision. Based on this exhaustive process, we identified knowledge, attitudes, self-efficacy, and LGBTQ+ affirming care as key criterion domains; these criterion domains are described in more detail in the parent study [26].”

On p. 128, what is "intrascale stochastic imputation" and is there reference citation associated in using that imputation method for missing data?

We have revised to describe this method, with citation (Methods, Missing Data and Data Quality):

“This method of imputation uses maximum likelihood estimation to impute missing items from non-missing items within internally consistent groups. It is appropriate when missing values occur randomly and there is sufficient internal consistency among items as we determined was consistent with  our missing data [27].”